

# Smart apiculture management services for developing countries—the case of SAMS project in Ethiopia and Indonesia

Kibebew Wakjira[1], Taye Negera[1], Aleksejs Zacepins[2], Armands Kviesis[2], Vitalijs Komasilovs[2], Sascha Fiedler[3], Sascha Kirchner[3], Oliver Hensel[3], Dwi Purnomo[4], Marlis Nawawi[4], Amanda Paramita[5], Okie Fauzi Rachman[5], Aditya Pratama[5], Nur Al Faizah[6], Markos Lemma[7], Stefanie Schaedlich[8], Angela Zur[8], Magdalena Sperl[8], Katrin Proschek[9], Kristina Gratzer[10] and Robert Brodschneider[10]

[1] Oromia Agricultural Research Institute, Holeta Bee Research Centre, Holeta, Ethiopia
[2] Latvia University of Life Sciences and Technologies, Jelgava, Latvia
[3] University of Kassel, Kassel, Germany
[4] University Padjadjaran, Sumedang, Indonesia
[5] Labtek Indie, Bandung, Indonesia
[6] The Local Enablers, Sumedang, Indonesia
[7] Iceaddis IT Consultancy PLC, Addis Ababa, Ethiopia
[8] Deutsche Gesellschaft für Internationale Zusammenarbeit (GIZ) GmbH, Feldafing, Germany
[9] Icebauhaus e.V., Weimar, Germany
[10] University of Graz, Graz, Austria

Corresponding author
Aleksejs Zacepins,
aleksejs.zacepins@llu.lv,
alzpostbox@gmail.com

## ABSTRACT

The European Union funded project SAMS (Smart Apiculture Management Services) enhances international cooperation of ICT (Information and Communication Technologies) and sustainable agriculture between EU and developing countries in pursuit of the EU commitment to the UN Sustainable Development Goal "End hunger, achieve food security and improved nutrition and promote sustainable agriculture". The project consortium comprises four partners from Europe (two from Germany, Austria, and Latvia) and two partners each from Ethiopia and Indonesia. Beekeeping with small-scale operations provides suitable innovation labs for the demonstration and dissemination of cost-effective and easy-to-use open source ICT applications in developing countries. SAMS allows active monitoring and remote sensing of bee colonies and beekeeping by developing an ICT solution supporting the management of bee health and bee productivity as well as a role model for effective international cooperation. By following the user centered design (UCD) approach, SAMS addresses requirements of end-user communities on beekeeping in developing countries, and includes findings in its technological improvements and adaptation as well as in innovative services and business creation based on advanced ICT and remote sensing technologies. SAMS enhances the production of bee products, creates jobs (particularly youths/women), triggers investments, and establishes knowledge exchange through networks and initiated partnerships.

## INTRODUCTION

Pollination through insects is basic to agricultural and horticultural plants. It has been estimated that 66% of the world's crop species are pollinated by a diverse spectrum of pollinators, including the polylectic honey bee (*Kremen, Williams & Thorp, 2002*; *Partap, 2011*). The symbiosis of pollinated species and pollinators is in a sensitive balance and the reduction and/or loss of either will affect the survival of both (*Abrol, 2011*; *Panday, 2015*). The pollination value was estimated to make up between 1 and 2 percent of the global GDP (*Lippert, Feuerbacher & Narjes, 2021*). Thus, the conservation of honey bees and other pollinators is of great interest to maintain biodiversity, to provide the world's food security, and in a broader sense to ensure our existence (*Potter et al., 2019*). The pollination process is crucial for the reproduction of cross-pollinated plant species, increases the yields and enhances their quality (*Fichtl & Adi, 1994*; *Eilers et al., 2011*; *Admasu et al., 2014*; *Klatt et al., 2014*). Besides the important aspect of pollination, honey bees also produce a variety of bee products, including honey, beeswax, pollen, royal jelly or propolis which also leads to an economic benefit for the beekeeper (*Crane, 1990*). Therefore, honey bees do not only play a key role in preserving our ecosystems, but also contribute to a greater income (*Bradbear, 2009*). During the last decade, honey bees got further into the center of the world's attention due to higher colony losses than usual (*Oldroyd, 2007*; *Van der Zee et al., 2012*; *Brodschneider et al., 2016*; *Brodschneider et al., 2018*; *Gray et al., 2019*; *Gray et al., 2020*). In 2007, the term colony collapse disorder (CCD) was coined for the depopulation of a honey bee colony (*Oldroyd, 2007*; *VanEngelsdorp et al., 2008*; *Dainat, VanEngelsdorp & Neumann, 2012*). The reasons for this phenomenon are not yet well understood, but it is suggested that proper hive management lowers the risk of CCD and colony losses (*Steinhauer, Van Engelsdorp & Saegerman, 2020*). Meanwhile, the role of bees for the world's economy and food security is undoubted and therefore not only scientists, but also farmers, ecologists, and policy makers join forces to make efforts in preserving them (*EFSA, 2013*).

Proper hive management and monitoring for pests, parasites, and diseases, as well as for colony strength, were identified to be crucial factors for honey bee health and productivity and therefore are regarded as vital elements of successful beekeeping (*EFSA, 2013*; *Steinhauer, Van Engelsdorp & Saegerman, 2020*). To assess those parameters, beekeepers must open the hive and visually inspect it regularly (*Van der Zee et al., 2012*; *Delaplane, Van Der Steen & Guzman-Novoa, 2013*). However, manual monitoring of beehives is a time-consuming process for beekeepers and stressful to bee colonies. Time-consumption even increases with the beekeeping sites' distance to the homesteads, so every inspection also incurs travel costs to beekeepers (*Meikle & Holst, 2015*; *Zetterman, 2018*). Further, honey bee species and subspecies differ in their behavior (*Gupta et al., 2014*). While the Asian honey bee *Apis cerana* is known for its gentle temperament and easy handling, African *Apis mellifera* subspecies are very aggressive, causing safety issues for the beekeepers during hive operation.

To facilitate the hive management procedure, the implementation of smart apiary management services is believed to be the future (*Bencsik et al., 2011*; *Edwards-Murphy et al., 2015*; *Meikle & Holst, 2015*; *Zacepins et al., 2016*). Differing from previous funded

European Union projects which focused mainly on European countries, SAMS (Smart Apiculture Management Services) received its funding under the specific purpose to target requirements of low and middle income countries in sub-Saharan Africa and ASEAN. In order to reach this goal, information and communication technology (ICT) tools based on remote sensing to monitor the bee colony's health and productivity are used (*Zacepins et al., 2015*). So far, several multi-dimensional monitoring information systems have been developed and applied in "Precision Beekeeping" (*Kviesis, Zacepins & Riders, 2015*; *Zacepins et al., 2015*; *Rodriguez et al., 2017*; *Komasilovs et al., 2019*; *Kontogiannis, 2019*), but only a few implemented solutions for honey bee data collection offer basic functionality for data analysis and decision making, and hence still need to be improved (*Kviesis, Zacepins & Riders, 2015*).

Precision beekeeping is increasingly implemented in Europe, but lags behind in Africa and Asia. The SAMS project focuses on beekeeping in Ethiopia (*Demisew, 2016*; *Negash & Greiling, 2017*; *Wakjira & Alemayehu, 2019*) and Indonesia (*Gratzer et al., 2019*) as in those countries a huge beekeeping potential is recognized but not unlocked yet.

A combined biological, sociological, and technical approach is made within the SAMS project. It enhances international cooperation of ICT and sustainable agriculture between the EU and developing countries to pursue the EU commitment to the UN Sustainable Development Goal "End hunger, achieve food security and improved nutrition and promote sustainable agriculture". The main objectives of SAMS are to develop, refine, and implement an open source remote sensing technology for monitoring the health and productivity of bee colonies. SAMS also aims to foster the regional added benefit and gender equality in employment. Furthermore, maintaining honey bees has a high potential to foster sustainable development also in other economic sectors, such as the beekeeping supply chain, forestry, agriculture or the beauty (cosmetics) sectors of developed and developing countries (*Bradbear, 2009*; *Gupta et al., 2014*). An important asset of this project is the co-creation of local systems to avoid falling into the same trap as other beekeeping programs in developing countries, like ignoring local skills and knowledge (*Schouten & Lloyd, 2019*). Furthermore, SAMS supports cooperation at international and national levels to promote mutual learning and research on open source technology, and best practice bee management for Africa and Asia.

This creates jobs, adds value to products and income, and hence contributes to the global fight against hunger (*Panday, 2015*; *Roffet-Salque et al., 2015*; *Patel et al., 2020*).

The aim of this paper is to give an overview of the SAMS project and present ideas and concepts that have been developed considering the needs and requirements of beekeepers, business facilitators, researchers and other stakeholders. The conceptual goals of SAMS and its methodology, which are based on the principles of User Centered Design (UCD) are introduced first, followed by a description of the developed standardized SAMS beehive, and hive monitoring system, which meet the needs of beekeepers in Indonesia and Ethiopia. Complementary to the SAMS hive monitoring system, insights on the developed data warehouse model to facilitate decision support for beekeepers, and SAMS activities, which support the sustainable growth of beekeeping, apiary construction businesses and the bee product market in these countries, are provided.

## Concept of the SAMS HIVE monitoring

Advanced ICT and remote sensing technologies enhance precision apiculture and help to increase the role of bees in pollination services as well as the production of hive products while maintaining a healthy environment. Precision apiculture is an apiary management strategy based on the monitoring of individual colonies without hive inspection to maximize the productivity of bees (*Zacepins et al., 2015*). Driven and based on the User Centered Design approach, SAMS is an apiary management service based on three pillars:

1. Development of modern and modular hives, adapted to the local context, equipped with a remote measurement system for bee colony behavior, productivity and health status monitoring,
2. Development of a cloud-based Decision Support System (DSS) to implement a management Advisory Support Service (ASS) for the beekeepers,
3. Development of adapted bee management guidelines about seasonal changes, available forage plants, and an ICT-data driven model for needed beekeeping actions.

## Human-centered Design (HCD) within SAMS

The whole process within SAMS followed a human-centered design approach (HCD), (*Deutsche Norm, 2019*). Human-centered design is a multi-step iterative process (see Fig. 1) which requires defined steps and includes understanding and analysing the context of use, specifying the user requirements, producing design solutions, and evaluating them against those user requirements, if possible, with user participation.

All actions and developments within the project were performed in close cooperation and collaboration with the end-users, especially with the focus user group: beekeepers.

A thorough user research and context of use analysis has been conducted to understand the preconditions of the local environment as well as the potentials and challenges for a successful technology supported apiculture. In order to understand beekeepers as SAMS focus users better, empirical methods like contextual interviews, observations, surveys, workshops, focus group discussions, and field studies have been undertaken. Results have been documented in the form of personas (https://wiki.sams-project.eu/index.php/Personas, last accessed: 18.02.2021) and as-is scenarios (https://wiki.sams-project.eu/index.php/AS-is_Scenarios, last accessed: 18.02.2021) and presented to all SAMS team members and beekeepers for review and refinement. Based on the review, the SAMS team and beekeepers identified and described user requirements and started a collaborative design thinking process to produce conceptual design solutions and low-level prototypes for essential products around the decision support system and the advisory support service for beekeepers. Those design solutions were iteratively evaluated and refined.

With the diverse contexts of implementation in Indonesia and Ethiopia, SAMS must meet the challenge of including culture specific variations in the prototyping process. These culture specific variations considered different beekeeping traditions, different bee types, and climate conditions as well as different languages, different social and political contexts. Multidisciplinary exchange of information and collaboration between local culture experts, beekeeping experts, hardware specialists, database architects, and software

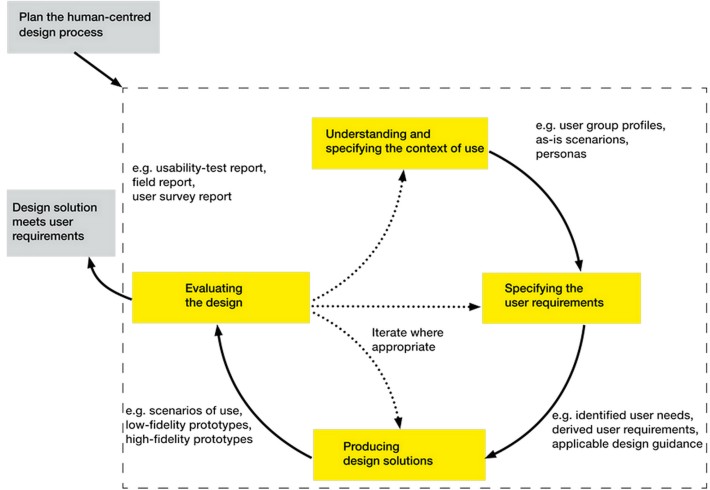

**Figure 1** **Human-centered design process as applied in SAMS project for development of interactive systems.** User participation drives the HCD process since in each iteration, the product design and context of use analysis steps are based on user feedback. Interdependence of human-centered design activities (*Deutsche Norm, 2019*).

engineering specialists were essential. The collaboration was motivated by a common goal to develop technically robust, reliable, easy-to-use, easy to maintain under the specific conditions and affordable services that provided added economic value to the beekeepers.

## Development and standard of SAMS beehive

One aspect of SAMS is to develop and standardize beekeeping practices within Ethiopia and Indonesia, respectively. To achieve this, the SAMS team constructed and developed a standard SAMS beehive, which can be used in future beekeeping and enables sensor placement and information technology implementation.

A modern beehive is an enclosed, man-made structure in which honey bee colonies of the genus *Apis* are kept for man's economic benefit (*Atkins, Grout & Dadant & Sons, 1975*; *Crane, 1990*). The design of such a hive should balance the requirements of the colony and convenience for the work of beekeepers. In traditional African hives, honey bees build their natural nest by constructing parallel combs vertically downwards from the roof of the nest cavity almost the same way as they do in wild nests. During comb construction, a space—called "bee space"—is left between the combs. Bee space, and comb spacing (midrib to midrib distances), and lots of other striking features are found to vary from species to species and among the different subspecies of a species (*Seeley, 1977*; *Jensen, 2007*). To gain insight into details of the requirements of honey bees, preliminary studies on bee space measurements from different agro-ecologies of Ethiopia and assessment of dimensions of different beehive components manufactured in different workshops have been conducted for *A. mellifera* colonies. For *A. cerana* requirements, different literatures were assessed and consulted, needs and requirements were analysed (*Jensen, 2007*; *Schouten, Lloyd & Lloyd, 2019*). The results from these studies were used in determining the bee space, comb spacing, and other hive dimensions to develop standards and material specifications for

new beehives according to the needs and nature of the two honey bee species targets by SAMS.

In selecting the prototype to design and develop a standard beehive for SAMS, various available prototypes have been considered. Improved modern beehives such as Langstroth, Dadant, Foam, Zander, and modified Zander have been assessed for their advantage and ease of construction. All of these prototypes were designed and optimized for *A. mellifera* and *A. cerana*. From the preliminary study and literature analysis, dimensions of different parts and procedures required for hive construction were carefully organized for the standard SAMS beehive so that a complete hive system can easily be produced locally and used in the beekeeping industry. For this purpose and the required criteria, Langstroth and its modified version, the Dadant model, were chosen for the standard SAMS beehive. The reasons for choosing these two prototypes were: (1) both hive systems have several hive boxes that can be stacked one above another to expand the hive volume, and have the possibility of confining the queen to the lowest chamber (brood box) by using a queen excluder; (2) familiarity of the hive systems in project countries and beyond. Almost all-commercial beekeeping operations throughout Europe, North America, Australia, and parts of South America and Asia and some African countries, operate based on the Langstroth and Dadant types (*Atkins, Grout & Dadant & Sons, 1975*; *Segeren & Mulder, 1997*). This universality can help to ease the adoption of the new SAMS beehive system among the beekeeping community, ensuring sustainability of the project; (3) these two beehive types can generate the highest honey yield, due to the option to add supers one above the other easily; (4) standardizing enables consistency of parts production across manufacturers in different workshops in different regions. This will bring hive parts prices down to reasonable levels and opens the opportunity to do business out of beehive production. Therefore, this can assure sustainability and create an impact on productivity and bee health, as this innovation can transform beekeeping activity into a full-scale industry.

The proposed beehive system is sketched in Fig. 2. The complete system consists of a loose bottom board, bottomless brood chamber, supers above brood chamber, inner cover, and outer cover. The bottom or lower chamber is used for the queen to lay eggs, and the supers serve as honey stores. The volume of each chamber is based on the assumption of 10 vertically hanging frames. Between the frames, other parts, and each frame, a bee space of 10 mm for *A. mellifera* and nine mm for *A. cerana*, allows movement of individual workers for comb construction, brood rearing, and storing food. However, the major difference in this development compared to previous prototypes is that the bottom board and inner cover are designed to serve additional purposes. The top part of the bottom board is covered by a wire grid with a $3 \times 3$ mm mesh size. The mesh allows debris to fall out of the beehive. The mesh floor also allows air circulation in the hive. From the rear side of the bottom board, a slot for placing a mite floor is created for the diagnosis of small arthropod pests like varroa mite, small hive beetle, or sugar ants. The mite floor contains a piece of waterproof plywood of similar size to the bottom area of the brood chamber. For pest control, any glue harmless to bees and products is smeared on the mite floor's upper side. The sticky materials then trap any pests. Another modification in the

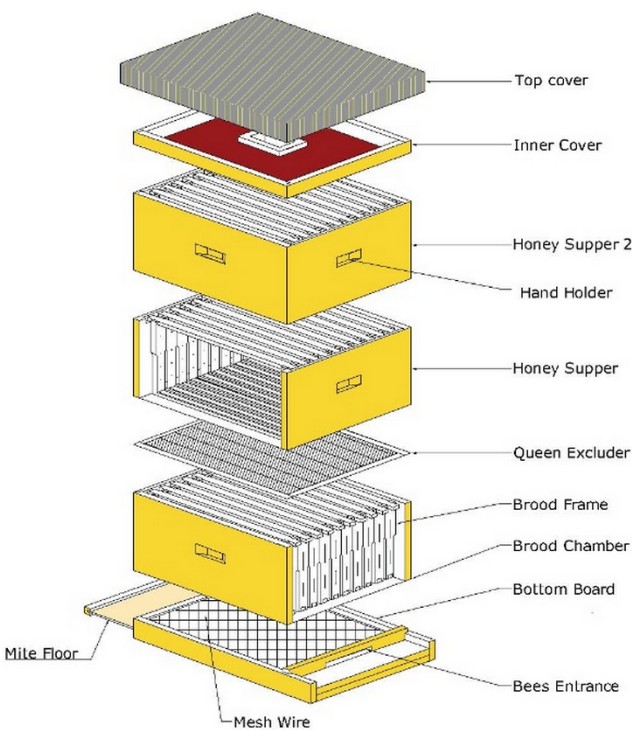

**Figure 2** **A complete proposed SAMS beehive system sketch.** Sketch describes all parts of the beehive - bottom board with bee entrance, brood chamber with frames, honey suppers and the top cover.

SAMS beehive is to fit the hive with an inner cover primarily used to cover the uppermost super before the outer cover. The inner cover is designed to prevent death of worker bees during hive operation due to breaking of propolis seal if only the outer cover is used. In this beehive system, the inner cover is designed to additionally serve as a feeder to supply bees with sugar syrup or pollen patties during dearth periods. Proposed dimensions and detailed views of the beehive bottom board is described in the SAMS manual on beehive construction and operation (https://wiki.sams-project.eu/index.php/Bee_Hive_Manual, last accessed: 18.02.2021).

## SAMS HIVE monitoring system

In modern beekeeping in Europe, precision beekeeping is well established with many commercial systems available for remote bee colony monitoring, mainly recording and transmitting weight measurements (*Lecocq et al., 2015*).

Some of these commercial solutions are expensive, and Ethiopian or Indonesian beekeepers cannot afford them. Some systems do not provide data transfer capabilities using mobile networks, and others do not work without a standard power supply. Thus, the SAMS HIVE monitoring system considers specifics of the two target countries and developing countries, based on the local beekeepers' needs.

The system contains several functional groups:

1. A power supply with a router to run up to 10 monitoring units;

2. A central computer unit where the sensors are connected;
3. A sensor frame placed in the beehive, including temperature and humidity sensor as well as a microphone;
4. A scale unit positioned beneath the beehive with an optional sensor for outdoor temperature and humidity monitoring.

The architecture diagram of the SAMS HIVE system is shown in Fig. 3. The power supply for the monitoring units is provided by a photovoltaic system (referred to as power unit) via cables. It consists of the standard components: solar module, charging controller, and battery. The power unit also supplies a mobile GSM Wi-Fi router, which is used as a hotspot for the monitoring units to transfer data to a web server (SAMS data warehouse).

The monitoring unit consists of a printed circuit board (PCB) with Raspberry Pi Zero W single-board computer, a step-down converter to change the voltage of the power unit to 5V, and a 24-bit analog-to-digital converter (ADC) that converts the Wheatstone bridge signals of the load cell to a digital format. The load cell measures the weight of the colony. The sensor frame with temperature and humidity sensor as well as a microphone is also connected to the computer. This module allows acoustic signals and colony parameters like temperature to be recorded. The acoustics are recorded over a certain timespan and uploaded as a Fast Fourier Transformed (FFT) spectrum and transferred to the SAMS data warehouse. It is recorded with 16 kHz sampling frequency, covering a frequency range from 0 kHz to 8 kHz. The FFT is made with 4096 points resulting in a frequency resolution of approximately 3.9 Hz.

The computer can be extended with additional sensors. For example, it is possible to connect a small weather station to collect region-specific climate data or additional temperature sensors to be placed in different hive locations (top, bottom, in frames). A deep sleep mode can be used in between the measuring intervals utilizing a power control unit (WittyPi) in order to reduce energy consumption considerably. As soon as the computer receives power from the power unit, it starts the measuring routine. The measuring routine and the interval can be adjusted remotely via online configuration as required.

After a successful recording, the data is transferred via Wi-Fi to the mobile GSM router and sent to the web server (Fig. 3). If the real time upload is not possible, the data remains on the SD card until a successful upload or remote collection has been performed. In this case, a new upload attempt starts after 30 s. Each device has its ID so that it can be uniquely assigned to the web server. Individual sensors can also be added to users, locations, or groups on the web server. Successful recording, data storage, uploads or errors are logged and transferred to the web server. Events for troubleshooting can be viewed there by administrators. On the device, 2 LEDs indicate working or deep sleep mode. Plug connections ensure easy installation. The sensor frame is connected to the computer via flat cable and IDC connectors. As a power supply connection, a standard DC power plug was selected. In addition to the sensor frame, a case was designed to place the monitoring unit's components. Both cases are 3D printable models (Figs. 4–6).

A software was developed to operate the Raspberry Pi and its components as a monitoring system. In order to ensure the simple and long-term availability of the code, a separate

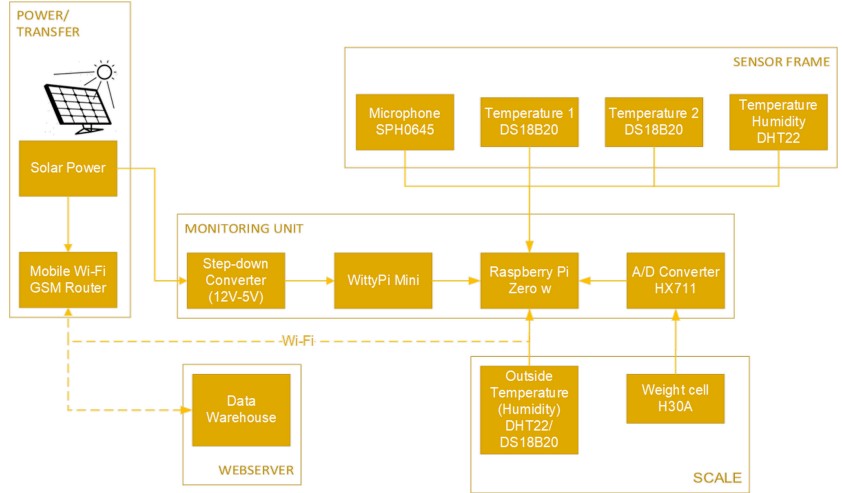

**Figure 3** **Flow chart of the SAMS HIVE system.** Power unit, scale unit, sensor frame and data warehouse.

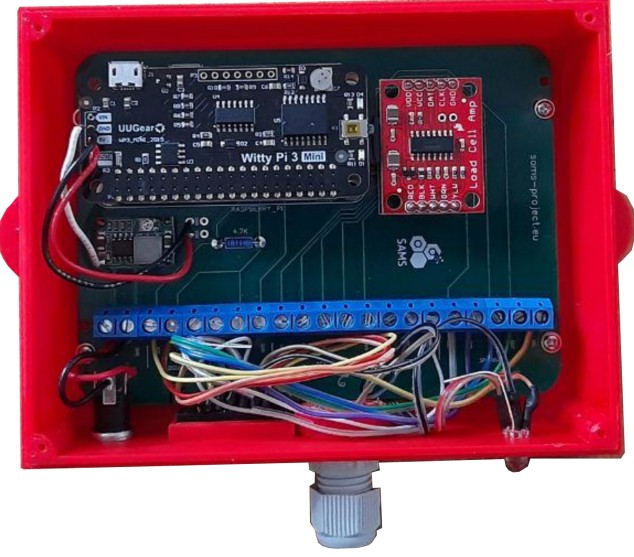

**Figure 4** **SAMS HIVE device.** Measurement device with ports and status LED.

SAMS page was created on the GitHub developer platform. The code (sams-app 2.47) is released under the MIT license and can be found at https://github.com/sams-project. The GitHub page contains the code to operate the monitoring system, a web application to calibrate the functions and the code to set up a data warehouse. Also, the files to print and build the PCB and cases are available there.

The recommended installation is to use a sensor frame placed in a brood frame (Fig. 7). The sensor frame is installed centrally in a brood frame so that the sensors are located in the middle of the brood nest.

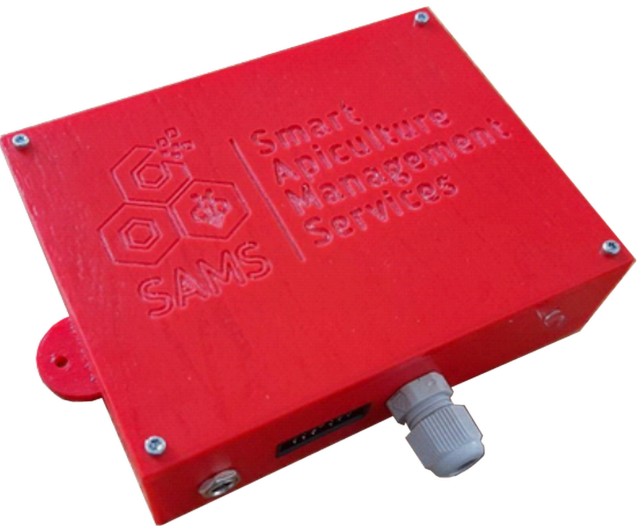

**Figure 5  SAMS HIVE case.** PCB and components placed in a 3D printable case.

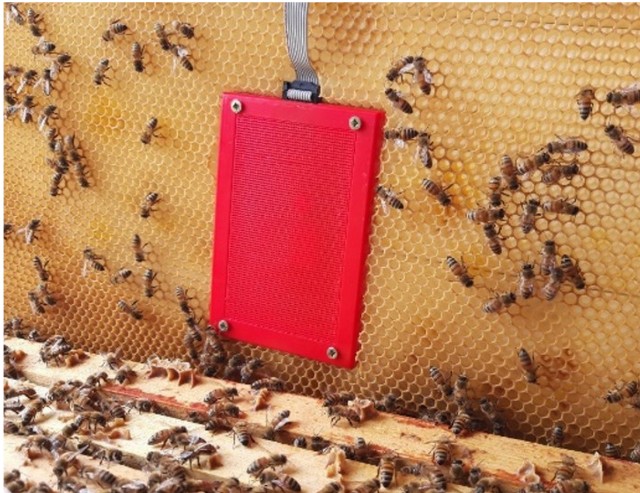

**Figure 6  SAMS HIVE sensor frame.** Sensors are installed in a 3D printable case placed in a regular brood frame and connected with flat cable to SAMS HIVE device.

The price of the SAMS HIVE monitoring system (current version 2) is about 170 \$. In addition, there are the expenses for power unit and GSM. The dimensioning of the photovoltaic system for the power unit depends on the location, the number of monitoring units and the measuring intervals. The cost of the photovoltaic system is about 200 \$ and up to ten monitoring units can be powered by it. Modular electronic components were used to ensure the sustainability of the monitoring system. The components can be replaced independently and also be used for other purposes. A recycling plan should support this if

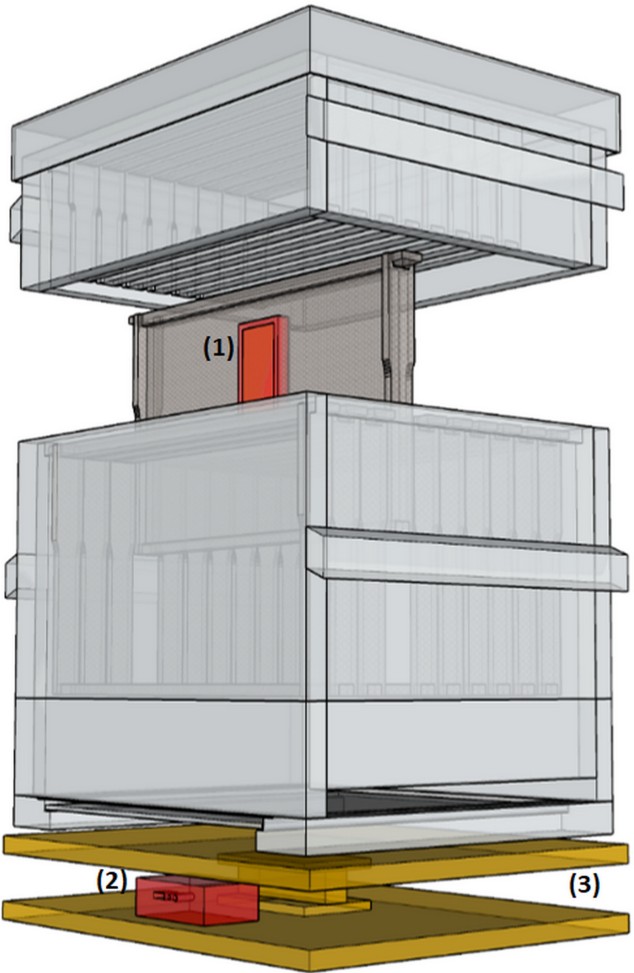

**Figure 7** **Placement of SAMS HIVE system.** Sketch of a common Dadant beehive with placement of: (1) Sensor frame in a brood frame, (2) HIVE case and (3) Scale unit.

necessary. In addition to its expandability, the system can also be set up for other academic and research applications and bee institutes to collect sensor data.

During UCD, implementation and testing of the SAMS Hive monitoring system, some observations were made. These observations (listed below) will significantly contribute to business potential mapping and development.

1. Beekeepers have a limited budget, and technology is not yet considered in beekeeping practices.
2. Local beekeepers found it valuable to monitor trap-hives (modern beehives used to trap new bee colony), placed deep in the forest, so power source became the main concern for such systems.
3. Cheaper monitoring system that is simple and easy to augment to the existing modern beehive is preferable.

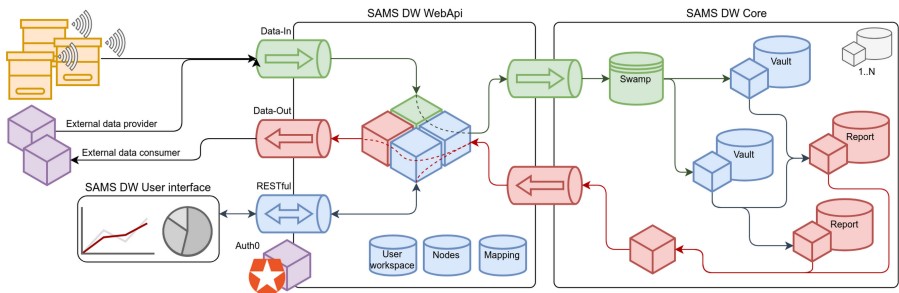

**Figure 8  Architecture of the developed SAMS data warehouse.** Main DW components are shown in frames (Core, WebApi, User interface). Cubes represent various processing units interacting with each other, cylinders represent persistent storage, pipes (horizontal cylinders) represent communication channels. Vaults and Reports in DW Core are independent processing units with dedicated storage (*Komasilovs et al., 2019*).

Some aspects concerning the beekeeping ecosystem in target countries also need to be considered; for example, the Indonesian beekeeping ecosystem is not yet as developed as the beekeeping ecosystem in Ethiopia or Europe. This immaturity of the ecosystem resulted in a lack of integrated support from beekeeping stakeholders. So simple technology is considered a better option first to improve the ecosystem.

## SAMS data warehouse and decision support system

All the measured data about the behavior of bee colonies, gathered from the HIVE monitoring system, can be stored for further analysis and decision support. For data storage, a dedicated data warehouse is developed (*Komasilovs et al., 2019*), which can be considered as a universal system and is able to operate with different data inputs and has flexible data processing algorithms (*Kviesis et al., 2020*). Architecture of the developed DW is demonstrated in Fig. 8. The DW is a fully operational solution, it is storing incoming data in real-time and is providing the infrastructure for the future data analysis, processing and visualization. The SAMS data warehouse is accessible by the link: https://sams.science.itf.llu.lv/. It is an open source software and it can be used by others to further extend its functionality, develop different user interfaces and/or native mobile applications, and use in new business opportunities. Data warehouse source code is accessible in the GitHub repository: https://github.com/sams-project. For the data analysis several approaches can be used, within the SAMS project a Decision Support System was implemented.

For the beekeepers the raw sensory data must be analyzed, interpreted and translated into clear instructions that consider the operational ability and beekeeping knowledge of the users. The main aim of the DSS is to detect and recognize various bee colony states (*Zacepins et al., 2015*) and inform the beekeeper about them. Still it needs to be noted that beekeepers remain as the final decision makers and can choose appropriate action and when to take it.

For the SAMS project each country context and environmental factors should be thoroughly analyzed to develop specific algorithms that allow safe interpretation. The

SAMS DSS has a modular design, consisting of a comprehensive expert interface, which has been developed and adapted together with local beekeepers and which can be used by apiculture experts, e.g., in a service and advisory support centers, to analyze and monitor data. Also, an easy to use and understandable application on smartphones or SMS service is required to alert beekeepers about hives that need attention. The user centered design approach allowed the technical layout and user interfaces to be developed in parallel, based on shared research results. Through the expert interface, local beekeeping experts can assist the beekeepers if needed. At this moment some of the models required for DSS are implemented into the SAMS data warehouse. A mockup of mobile application interface was created according to local user needs and is publicly available, allowing further development by interested parties.

## Api-management within SAMS

Api-management is central to the SAMS project, including the contextualizing of local systems focusing on the two target countries Ethiopia and Indonesia, the development of an open source and agile database and a honey bee health and management related capacity building strategy. Even though Europe's beekeeping sector is comparably strong, it relies on honey imports from third world countries as its production is not sufficient enough to saturate the market (*García, 2018*). While governmental involvement and subsidized national programs aim to strengthen the stagnated European bee product market, such programs are lacked completely in Indonesia (*Gratzer et al., 2019*), and are not carried out sustainably enough to set the beekeeping sector of Ethiopia on a par with those of other global players. In Europe, beekeeping has a long tradition and knowledge is accessible by numerous books and journals. Bee health is affected by a diverse spectrum of organisms (protozoa, fungi, bacteria, insects, mites, etc.) (*Bailey & Ball, 1991*; *Generesch, 2010*), but the parasitic mite *Varroa destructor*, introduced to Europe, is the major threat to European honey bees (*Rosenkranz, Aumeier & Ziegelmann, 2010*). The varroa mite seems to be no big issue for Ethiopian nor for Indonesian honey bees but this is not well documented. However, several other organisms affect Ethiopia's bees, including protozoa, fungi, insects, birds and mammals, but with the exception of ants or wax moths, mostly no control methods are applied (*Ellis & Munn, 2005*); Awraris Getachew (*Shenkute et al., 2012*; *Tesfay, 2014*; *Pirk et al., 2016*).

In Ethiopia, beekeeping dates back ∼5000 years (*Tekle & Ababor, 2018*), and more than one million households maintain around six million honey bee (*A. mellifera*) colonies producing more than 50,000 tons of honey per year, making Ethiopia Africa's leading honey and beeswax producer (*Degu & Megerssa, 2020*). However, Ethiopia's honey sector is far behind its potential of 500,000 tons per year. The reasons include limited access to modern beekeeping practices and equipment, a shortage of trained people, the use of agriculture chemicals, the impact of droughts, absconding and the lack of infrastructure and market facilities (*Yirga et al., 2012*; *Legesse, 2014*; *Fikru & Gebresilassie, 2015*; *Degu & Megerss, 2020*). The vast majority of hive systems in Ethiopia are traditional, some are classified transitional (top bar hives), only few are classified as modern hives. Traditional hives are made from locally available, but often non-durable materials (clay, straw, bamboo,

logs, etc.). Even though this kind of hive system requires low starting costs and skills, honey harvesting is always accompanied by destroying large parts of the bees' nest. Furthermore, the productivity is considered to be low (*Yirga & Teferi, 2010*; *Beyene et al., 2015*; *Degu & Megerssa, 2020*). Traditionally, beekeepers gain their knowledge from the family or village (*Fichtl & Adi, 1994*). As training centers are rare in Ethiopia and beekeepers from rural regions often lack infrastructure, access to modern beekeeping knowledge and techniques is restricted. One of the largest bee research institutions in the country is a one hour drive away from the capital Addis Ababa. The Holeta bee research center is involved in educating beekeepers and connecting them by offering training and hard copies of training manuals for beginners and advanced beekeepers including now the SAMS manual for beekeeping equipment production.

So far, classic beekeeping training centers do not exist in Indonesia. To be able to establish one, one must face political and social issues first as the awareness of the importance of bees for the ecosystem was reported to be low in the country. Furthermore, in relation to the large Indonesian population size, beekeeping is not widespread and beekeeping-related literature is not readily available (*Gratzer et al., 2019*). Honey hunting has tradition in parts of the country, but managing honey bees in hives is a comparatively young activity in Indonesia. Most beekeepers keep the native Asian honey bee *A. cerana*, followed by the introduced *A. mellifera* which is mainly used for migratory beekeeping. While *A. cerana* is regarded less productive than *A. mellifera*, it is known for its easy handling and gentle behavior. One major problem identified, similar to Ethiopia, is the absconding behavior of bees. During unfavorable conditions, the colonies leave their hives, resulting in financial losses for beekeepers. Although many reasons for the underdeveloped beekeeping sector overlap with those of Ethiopia, others are specific to Indonesia, such as a lack of quality standards for bee products (*Crane, 1990*; *Masterpole et al., 2019*). Overall, there has been a sharp increase in beekeeping development publications over the past five years, but compared to Sub-Saharan Africa, the absolute number of publications for South Asia including Indonesia is rather low (*Schouten, 2021*). Due to the limited access and availability of literature, little information is given on bee health issues, control methods or management of honey bees in Indonesia, and therefore more research and lobbying efforts are highly recommended (*Gratzer et al., 2019*). As contextualizing is an ongoing process, an open source knowledge database was developed - the "SAMSwiki" (https://wiki.sams-project.eu, last accessed 18.02.2021). During the set-up, the SAMSwiki was fed with more than 200 literature sources including a variety of beekeeping related topics like Indonesian and Ethiopian bee sector parameters, bee forage, management options, bee health, as well as funding opportunities for businesses and SAMS-system related content. With its wiki-like approach, the readers can easily become members and contributors and are able to share their expertise with the remaining community. Extension of this database to other countries is planned for the future.

## Possibilities for smart bee management

Managed honey bee colonies need regular monitoring actions. Especially during the active foraging season, external and internal hive inspection is a necessary task for each

beekeeper. Those actions are time-consuming and regular opening of the beehive is a stress factor for the whole colony. With smart management, or precision beekeeping, those mandatory interferences are reduced to a minimum (*Bencsik et al., 2011*; *Meikle & Holst, 2015*; *Zacepins et al., 2015*). Smart bee management possibilities can be manifold and some of them, including the most relevant ones for the SAMS-project, are represented in Table 1. We elaborated what-if scenarios for the four most important events. For example, the start of a mass nectar flow indicates honey yield in the near future and beekeepers estimate this event either by knowing the vegetation in the surroundings by observing the flight entrance or by checking the food stores inside the hive; but a technical solution would make the beekeepers' work more efficient. Easy to understand illustrations have been developed for each important bee colony state, including basic recommendations for the beekeepers. One example can be seen in Fig. 9. The beekeeper gets informed as soon as an increase in weight of the monitored beehive by a certain, prior defined, percentage-value occurs. On detection of this event, further actions can be planned without even being present at the apiary. A typical event occurring only in African or Asian colonies is absconding, which has not been studied before using precision beekeeping approaches.

## Business models within SAMS

In addition to the open source remote sensing technology for monitoring the health and productivity of bee colonies, SAMS fosters the regional added benefit by identifying business opportunities and challenges, supporting business model development and thus assisting job creation. Enabling the SAMS team to identify SAMS business models several methods such as co-creation, ideation and observation of existing businesses were used. Ethiopia with its great potential in the apiculture sector has a wider range of business compared to Indonesia, and mainly focuses on beekeeping management. There are only a few businesses that offer derivative products, while Indonesia has only a few businesses that could improve beekeeping management as well as technology-based business.

One aspect became very clear during this project sequence—business development in the apiculture sector depends on the country readiness. Several factors indicate this country readiness, e.g., the maturity of the apiculture industry, government support, and age structure (children and young adolescents, the working-age population, and the elderly population). The more mature the apiculture sector in one country, the bigger the support given by the government, the more resources flow, the more flourishing the industry will be. The bigger the working-age population in one country, the more labor is available, the more industries are thriving. The working-age population factor is believed as one of the main factors that determine the growth of the creative industry. In 2018, the working-age population in Ethiopia was 55.26%, in Indonesia 67.59%, and in EU 64.69%.

As one of the SAMS goals is to provide a platform for concepts and ideas for local business developments, in order to have a sustainable long-term impact, an overall concept of SAMS business models was created and main obstacles in Ethiopia and Indonesia were identified. The 54 identified SAMS business models are rated based on its correlation to SAMS objectives and are recognized as SAMS business models that contribute in giving added value to the project aims and impact. All SAMS business models remain freely available

**Wakjira et al. (2021),** *PeerJ Comput. Sci.,* **DOI 10.7717/peerj-cs.484**

**Table 1  Ranking of smart management possibilities for bee colony state detection in Ethiopia and Indonesia.** Bold events/states were identified to be most relevant for the SAMS project. Asterisks (*) rank the importance, technical feasibility, grade of innovation and predictability of each event or colony state.

| Event or State of the colony/hive | Importance to the beekeeper (from less* to most important***) | Traditional detection methods | Parameter to measure | Technical feasibility (from easy* to complicated***) | Innovation (from already existing* to new***) | Predictability (not⁻ or from easy* to complicated***) |
|---|---|---|---|---|---|---|
| **Absconding** | *** | Detection after event happened | Temp., weight | * | *** | – |
| **Death** | *** | Internal and external inspection of the hive | Temp., sound, weight | * | * | – |
| **Start of the mass nectar flow** | *** | Observation of the flight activity outside the hive; internal inspection of the hive | Weight | * | * | Flowering calendar |
| **Broodless** | **(*) | External and internal inspection of the hive | Temp., sound | ** | ** | – |
| Queenless | **(*) | Internal inspection of the hive | Temp., sound | *** | ** | – |
| Colony Collapse | ** | Detection after event happened | Temp., weight | * | *** | – |
| End of the nectar flow | ** | Internal inspection of the hive; observation of the surrounding environment (flowers in bloom) | Weight | ** | * | ** |
| Pre-Swarming | ** | Internal and external inspection of the hive | Sound | *** | *** | – |
| Swarming | ** | Detection of the swarmed colony (after event happened) | Temperature, sound, weight | *** | ** | *** |
| Colonisation of an empty hive | ? | External and internal inspection of the hive | Temp., sound, weight | * | *** | – |

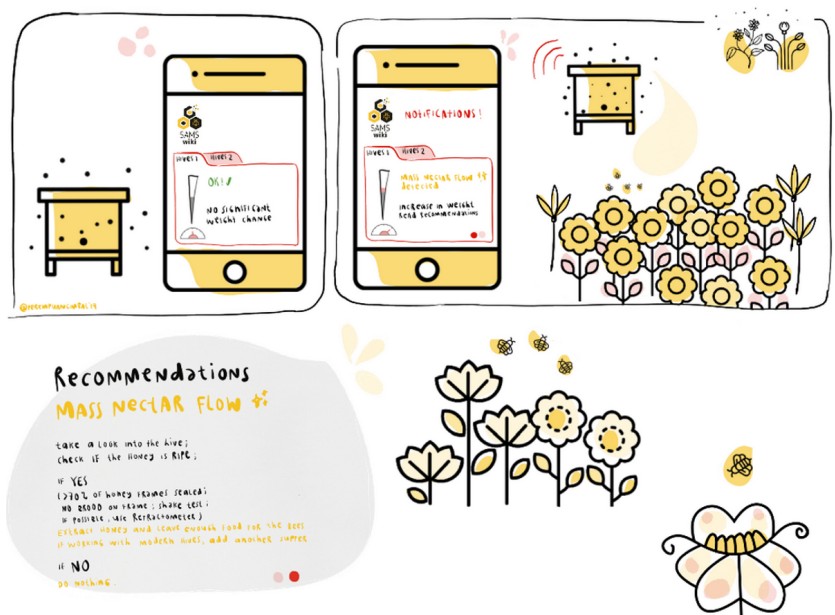

**Figure 9** **Exemplary illustration of the nectar flow as one smart bee management possibility.** Mass nectar flow is detected by the SAMS hive monitoring and decision support system, which triggers an alert on smartphones and recommendations for beekeepers.

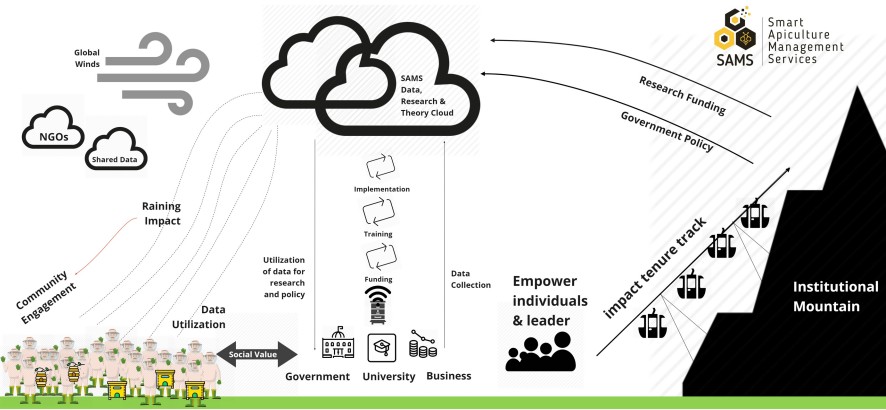

**Figure 10** **Overall concept of the SAMS business model.** Collaboration between government, university and business for achieving the specific goals is demonstrated in the concept.

on the SAMSwiki (https://wiki.sams-project.eu/index.php/SAMS_-_Business_Models https://wiki.sams-project.eu/index.php/SAMS_-_Business_Models, last accessed on 18.02.2021) also after the project end to enable stakeholders around the world to take up SAMS ideas and business concepts and to better position the apiculture sector in their own countries. Figure 10 illustrates the overall concept of the SAMS business models that involves various stakeholders in the process.

SAMS has a wider impact on the development of honey bee businesses by involving various stakeholders during the SAMS development and contextualization. The SAMS data, research and theory cloud represents all the knowledge acquired and collected during the SAMS project.

SAMS technology produced from the research process aims to make beekeeping activities more effective and efficient. To implement this product to its beneficiary, namely beekeepers, the high costs of its production makes it difficult to promote it directly, unless funding schemes from collaborations between government and business people and research institutions/universities are considered.

SAMS data that is utilized by the government (described as institution mountain), is useful for policy making in the fields of forestry, animal husbandry, agriculture, and the environment. The policy is then derived as an intake of community empowerment, leaders and other driving nodes. This concept is also expected to provide valuable benefits for the stakeholders involved. For beekeepers, bee colony management technology (SAMS) developed is obtained free of charge, as well as raising awareness in protecting the environment and government policies that support beekeepers and environmental communities. For governments, universities and businesses as funders, getting data from the technology applied to the colonies maintained by beekeepers for research and policy making.

Three main directions have strong impact on the SAMS ecosystem:

1. Practice - Individuals play a key role in driving institutional changes and therefore were identified as important for the SAMS ecosystem. Therefore, it is of major importance to recognize key individuals amongst a larger group of potentials and further empower them.

2. Institutional - International partnerships were initiated to support the SAMS ecosystem on business development, bee colony data and knowledge exchange, apiculture technology and services. Furthermore, the SAMS technology enables social innovation to engage more socially aspirational younger generations (i.e., their customers) to be more involved in the honey and bee industry.

3. Systemic - Social issues have an impact on the SAMS technology application in Indonesia. The market survey supported the research by mapping participant survey responses including all respondent-identified potentials in supporting the future business model of SAMS application. Wealth was also identified in the interviews as a key determinant of all identified issues. How to develop SAMS businesses and maintain their sustainability showing the interrelated nature of technology and also social problems, reinforcing the need for a collaborative, multi-agency approach to overcome the challenges in implementing the SAMS technology.

## CONCLUSIONS

The SAMS project developed an open source information and communication technology that allows active monitoring and managing of bee colonies to ensure bee health and bee productivity. For the first time, focus was given to special conditions of Africa and Asia,

including thorough research on actual user needs. Continuous monitoring of variables associated with honey bee colonies, including weight changes, temperature, humidity, acoustics, activity at entrance for detection of different bee colony states like swarming, broodless stage, and others becomes feasible for most practical applications. Established European or North American systems are not designed for the peculiarities that can be expected when monitoring colonies in Africa or Asia. Application of the SAMS design process allows the requirements of beekeeping in different countries and settings to be met, enhancing sustainable agriculture worldwide. To develop SAMS for local contexts, the project collected data from different user groups (individual beekeepers, beekeeping cooperatives, private and public input suppliers like beehive producers, beekeeping experts and researchers, and others) within the UCD process and enabled the team to adapt the system to specific requirements. At the end of the project, a greater awareness will be created in Indonesia and Ethiopia in regard to beekeeping and its activities and opportunities for greater income. There will also be the possibility to use collected data from different regions to better understand the behavior of bees and the environmental aspect and to ensure food production and bee farming activities. In addition, an international partnership network will ensure knowledge exchange and mutual learning.

Main results of the SAMS project are: (a) a manual for the SAMS monitoring beehive model, that is locally produced and adapted to local conditions, including integrated open source sensor and information transition technology, as well as an energy-supply solution; (b) the SAMS data warehouse which can be individually adapted; (c) a decision support system interface that can combine the sensor-based data-outputs with other information sources and predictive models to measure, analyze and describe different states of the bee colony such as health, vitality and production, (d) the SAMSwiki which provides knowledge on beekeeping in Ethiopia and Indonesia but also for other regions and (e) 54 SAMS business models for greater income opportunities and related upscaling potential.

### Funding
The SAMS project 'Smart Apiculture Management Services' and this publication received funding from the European Union's Horizon 2020 research and innovation programme under Grant Agreement No 780755. The funders had no role in study design, data collection and analysis, decision to publish, or preparation of the manuscript.

### Grant Disclosures
The following grant information was disclosed by the authors:
The European Union's Horizon 2020 research and innovation programme: 780755.

### Competing Interests
Markos Lemma is employed by Iceaddis IT Consultancy PLC, Stefanie Schädlich, Angela Zur and Magdalena Sperl are employed by GIZ, Amanda Paramita, Okie Fauzi Rachman and Aditya Pratama are employed by Labtek Indie and Katrin Proschek is employed by Icebauhaus e.V., Nur Al Faizah is employed by the Local Enablers.

## Author Contributions

- Kibebew Wakjira, Dwi Purnomo, Marlis Nawawi and Amanda Paramita conceived and designed the experiments, performed the experiments, prepared figures and/or tables, authored or reviewed drafts of the paper, and approved the final draft.
- Taye Negera and Oliver Hensel conceived and designed the experiments, authored or reviewed drafts of the paper, and approved the final draft.
- Aleksejs Zacepins conceived and designed the experiments, performed the experiments, analyzed the data, prepared figures and/or tables, authored or reviewed drafts of the paper, and approved the final draft.
- Armands Kviesis, Vitalijs Komasilovs and Sascha Fiedler conceived and designed the experiments, performed the experiments, analyzed the data, performed the computation work, prepared figures and/or tables, authored or reviewed drafts of the paper, and approved the final draft.
- Sascha Kirchner and Robert Brodschneider conceived and designed the experiments, analyzed the data, authored or reviewed drafts of the paper, and approved the final draft.
- Okie Fauzi Rachman conceived and designed the experiments, performed the experiments, authored or reviewed drafts of the paper, and approved the final draft.
- Aditya Pratama, Markos Lemma, Stefanie Schaedlich, Angela Zur and Magdalena Sperl conceived and designed the experiments, authored or reviewed drafts of the paper, and approved the final draft.
- Nur Al Faizah conceived and designed the experiments, prepared figures and/or tables, and approved the final draft.
- Katrin Proschek conceived and designed the experiments, prepared figures and/or tables, authored or reviewed drafts of the paper, and approved the final draft.
- Kristina Gratzer conceived and designed the experiments, analyzed the data, prepared figures and/or tables, authored or reviewed drafts of the paper, and approved the final draft.

## Data Availability

Software for running a Raspberry Pi as a data logger for sensor data on temperature, humidity, acoustics and weight is available at GitHub: https://github.com/sams-project/sams-app.

Software includes configuration of measurement intervals and frequency spectra as well as calibration of the load cell to log weight data.

This software can only be used in conjunction with a Raspbian Stretch Image (https://www.raspberrypi.org) and is developed within the SAMS project.

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
