# Peer review of "Smart apiculture management services for developing countries—the case of SAMS project in Ethiopia and Indonesia"

_PeerJ Computer Science, doi:10.7717/peerj-cs.484_

## Round 0.1 · original submission · Major Revisions

Firstly, I must apologise for the delay in returning these reviewer comments to you. I was pleased to see that both reviewers have welcomed your manuscript, and R2 in particular offers a number of minor revision suggestions. I also found the manuscript very informative, and clearly indicates that a considerable amount of work has been accomplished by the SAMS project stakeholders.

Unfortunately, I have noted several areas where the text of the manuscript need revision in order to improve both the paper's clarity and rigor.

1. Please take a look at the attached PDF where I have provided suggested rewordings, and noted sections that should be rewritten or restructured (particularly the final sections describing the SAMS Business Model and Conclusions sections). If possible, I recommend a proof reader with excellent written English is consulted prior to submitting your revised manuscript.

2. Many of the figures lacked clear titles and legends describing their essential message. I have added notes explicitly to some, but please also refer to the instructions to authors in this regard https://peerj.com/about/author-instructions/cs#figures

3. Please ensure all code, design documents and data files cited in the paper are properly versioned (e.g. tagged in a git repository and/or uploaded to Zenodo https://zenodo.org/, or given a versioned URI hosted on SAMS own servers). Where data or statistics have been obtained from other sources (e.g. FAOSTAT - http://www.fao.org/faostat/en/#home) please give date of access and a specific URL to allow others to view the data at its source. Data may also be downloaded and archived (e.g. via Figshare or Zenodo) if a specific URL cannot be obtained.

4. Clearly and consistently indicate the progress of each part of the project - e.g. the Data warehouse is variably described in the text as 'proposed', and a smart phone app is mentioned but no further details are presented. Ongoing work and future plans are certainly relevant for a paper like this, particularly if there is an opportunity for others to contribute (e.g. through open source software development), so please consider highlighting such further work in the final section of the paper.

Reviewer 1 ·

Basic reporting

I really enjoyed this article. It is well done and clearly presented. The last paragraph of text in the conclusion section seemed to have some formatting issues, but otherwise did not see any issues.

Experimental design

This was a case study, so there was not much in the way of experimental design. The project was presented and explained well and clearly, including acknowledging and address potential weaknesses.

Validity of the findings

Very nice as a case study. Hard to question the validity of their expierence.

Additional comments

I very much enjoyed reading this article.

Reviewer 2 ·

Basic reporting

The paper presents a useful tool for developing beekeeping solutions.
The process of design/development was well described and defined.
Moreover, the authors also proposed a standard beehive, hopefully, useful for the two target countries. Other strengths of the work:
- Clear and unambiguous, professional English used throughout.
- Literature references, sufficient field background/context provided.
- Professional article structure, figures, tables.
Work weakness:
- Lack of field experimentation.

Experimental design

Strengths of the work related to experimental design:
- Original primary research within the aims and scope of the journal.
- Methods described with sufficient detail.
Work weakness
The authors must define a research question.

Validity of the findings

Conclusions are well stated, but not linked to the research question.

Additional comments

All text - Put the names of species os bee highlighted. Preferably use italics.
Line 58 - Please verify the Eva Crane format citation.
Line 111 - Witch other?
line 113 - API management.
Line 137 to 142 - please avoid the use of quotations.
figure 1 - the 'design solution' shouldn't be before 'producing solution'?
line 300 - Is it 200 € for each monitoring unit?
line 308 - Should be stored or was stored.
In figure 6 - what means the blue and red cubes in this figure? Please insert a subtitle for each element.
line 355 - Around 95% of ...
lines 478 to 484 - a problem in the draft formation.

---

## Round 0.2 · Minor Revisions

Thank you for submitting your revised manuscript. I appreciate that the previous round included substantial revisions, and I was pleased to see that your revised manuscript was very much improved. In particular, the reordered Events in Table 1 makes it now extremely clear, and I also appreciate the work you have put in to improve and include additional figures.

I apologise for the delay in providing my editorial decision. One reviewer commented on your manuscript and found it acceptable for publication, but I note below a number of issues with the text that still need to be addressed before the manuscript is ready for publication. I think it is also worth mentioning that this kind of paper - which combines both social, governmental, societal, technological and ecological work is very tricky to write, but the SAMS project work you present is of sufficient quality that it should be represented by an equally high-quality academic paper.

Please find below my recommended revisions. Most are small, but there are some more problematic issues concerning the description of the SAMS business model, and the legends of certain figures. As I recommended previously - many of these changes are simply to fix English language grammatical issues and improve the clarity of the text, and I recommend you consult a colleague with excellent written and spoken English to help you to address these issues.


line 146: missing closing bracket for the HCD reference.

line 181: italicise 'Apis'

line 212: suggest: "universality can help easy adoption of the.."

line 257-258: relocate optional to say: "with an 'optional' outdoor temperature.."

line 259-266 and Figure 3. Suggest you relocate the reference to Figure 3 to the beginning of this section, and add 'referred to as' before 'power unit'. Please also be consistent later in its use (e.g. in line 303 you mention additional expenses for the "power supply" - this should be 'power unit' instead).

e.g.:
The flow chart of the SAMS HIVE system is shown in Figure 3. The power supply for the monitoring units is provided by a photovoltaic system (referred to as power unit) via cables. It consists of the standard components: solar module, charging controller, and battery. The power unit also supplies a mobile GSM Wi-Fi router, which is used as a hotspot for the monitoring units to transfer data to a web server (SAMS data warehouse).

NB. Figure 3 is not really a 'Flow Chart' - suggest instead you call it an 'Architecture Diagram'.

line 311-314. change 'Findings' to 'Observations' and Fix grammar. Suggest revision to:
"During UCD, implementation and testing of the SAMS Hive monitoring system, some observations were made. These observations (listed below) will significantly contribute to business potential mapping and development:"

line 323: insert "as" so it reads '..is not yet *as* developed as..'
line 330: remove 'the', add a comma and 'a': "For data storage, a dedicated"
line 342-347. You seem to repeat the aims of the DSS twice here. Suggest omitting line 342-345 since wording in line 345-347 is more concise.

line 349-line 360 is quite garbled. Suggest revising like:
"The SAMS DSS has a modular design, consisting of a comprehensive expert interface, which has been developed and adapted together with local beekeepers and which can be used by apiculture experts, e.g. in a service and advisory support centers, to analyze and monitor data. Also, an easy to use and understandable applications on smartphones or SMS services is also required to alert beekeepers about hives that need attention. The user centred design approach allowed the technical layout and user interfaces to be developed in parallel, based on shared research results. Through the expert interface, local beekeeping experts can assist the beekeepers if needed. At this moment some of the models required for DSS are implemented into the SAMS data warehouse. A mockup of mobile application interface was created according to local user needs and is publicly available, allowing further development by interested parties."

line 367: insert 'world': 'third world countries'

line 369: 'such programs are lacked completely in Indonesia' - or 'are completely lacking in Indonesia'

line 396-397: suggest '.. often lack infrastructure, access to modern beekeeping knowledge and techniques is restricted."

line 446: 'which has not been studied before using precision beekeeping approach.' - suggest 'approaches'

line 453: 'ideathlon' is not a word - suggest it was 'ideation'

line 456: insert "a" - ie 'only a few' in "only few businesses that"

line 476: " to create a greater position of the apiculture sector" - suggest 'to better position the apiculture sector'

Lines 478-503: After starting out well, the 'Business models within SAMS' section becomes very disconnected and hard to follow. I can suggest some rewordings, e.g.
line: 480-483: this paragraph doesn't make sense: "In the context of SAMS, the ecosystem pattern can be developed and contextualized in the development of SAMS which is directed to have a wider impact on the development of honey bee businesses. Ecosystem is developed by involving various stakeholders who carry out their respective roles." What is the 'ecosystem' here ? it might be best to put this more simply, e.g. 'SAMS can have a wider impact on the development of honey bee businesses by involving various stakeholders during SAMS development and contextualization.'

Line 484-503 seem to be a mixture of brief sentences and descriptive text. Some of that text that could instead be located in Figure 10's legend (e.g. line 491-493: 'raining impact'), others need to be revised and made more coherent. Line 504-523 are quite clear, but I am not sure what 'concept' these three points are supposed to be supporting.

Line 538: 'the UCD processed' should be 'the UCD process'.

Line 547: insert 'an' before energy: 'as well as [an] energy supply solution'

line 552: "e) 54 SAMS business models' - it looks like '54' is a typo, or do you really mean that there are 54 business models provided by SAMSwiki ?

Figure 1: "Significant is the user participation in this process, iterations of product design as well as iterations of context of use analysis are driven by user feedback. Interdependence of human- centred design activities [ISO /FDIS 9241-210:2019]."
- this legend needs to be reworded to be grammatically correct and easy to read.

Figure 9: "Exemplary illustration of the nectar flow as one smart bee management possibility
Mass nectar flow is detected by the SAMS hive monitoring and decision support system, which triggers an alert on smartphones and recommendations for beekeepers." suggest slight revision to "Example illustration showing the use of the smart bee management system. Mass nectar flow is detected by the SAMS hive monitoring and decision support system, which triggers an alert on smartphones along with recommendations for beekeepers."


Figure 10. You should relocate the descriptive text in lines 484-503 to here, in order to help the reader interpret this illustration. Please also address typos:
- 'intitutional mountain' should be 'institutional mountain'.

Reviewer 2 ·

Basic reporting

The authors did a good work taking into account all comments and suggestions by the reviewers. I have no further comments for the Authors.

Experimental design

The authors did a good work taking into account all comments and suggestions by the reviewers. I have no further comments for the Authors.

Validity of the findings

The authors did a good work taking into account all comments and suggestions by the reviewers. I have no further comments for the Authors.

Additional comments

The authors did a good work taking into account all comments and suggestions by the reviewers.
I have no further comments for the Authors.

---

## Round 0.3 · accepted · Accept

Thank you for addressing my requested revisions in the previous iteration. When preparing the final version of the manuscript please ensure the following are addressed:

1. Line 296 - reword and include license: "The code (sams-app 2.47) is released under the MIT license and can be found at https://github.com/sams-project";
2. Line 357 - Add full stop at end of sentence.
3. I assume that the final version of figure legends is as included in main text lines 735-776 rather than those appended in the review PDF (52718-v2). All the figure and table legends are satisfactory apart from Figure 1's title and legend, which should be revised to:

"Human Centred Design Process as applied in SAMS project for development of interactive systems. User participation drives the HCD process since in each iteration, the product design and context of use analysis steps are based on user feedback."

I look forward to seeing further outcomes and continued impact from the SAMS project in the future!